# AMiGA: Software for Automated Analysis of Microbial Growth Assays

Firas S. Midani,[a,b] James Collins,[a,b*] Robert A. Britton[a,b]

aDepartment of Molecular Virology and Microbiology, Baylor College of Medicine, Houston, Texas, USA
bAlkek Center for Metagenomics and Microbiome Research, Baylor College of Medicine, Houston, Texas, USA

**ABSTRACT** The analysis of microbial growth is one of the central methods in the field of microbiology. Microbial growth dynamics can be characterized by meaningful parameters, including carrying capacity, exponential growth rate, and growth lag. However, microbial assays with clinical isolates, fastidious organisms, or microbes under stress often produce atypical growth shapes that do not follow the classical microbial growth pattern. Here, we introduce the analysis of microbial growth assays (AMiGA) software, which streamlines the analysis of growth curves without any assumptions about their shapes. AMiGA can pool replicates of growth curves and infer summary statistics for biologically meaningful growth parameters. In addition, AMiGA can quantify death phases and characterize diauxic shifts. It can also statistically test for differential growth under distinct experimental conditions. Altogether, AMiGA streamlines the organization, analysis, and visualization of microbial growth assays.

**IMPORTANCE** Our current understanding of microbial physiology relies on the simple method of measuring microbial populations' sizes over time and under different conditions. Many advances have increased the throughput of those assays and enabled the study of nonlab-adapted microbes under diverse conditions that widely affect their growth dynamics. Our software provides an all-in-one tool for estimating the growth parameters of microbial cultures and testing for differential growth in a high-throughput and user-friendly fashion without any underlying assumptions about how microbes respond to their growth conditions.

**KEYWORDS** computational biology, growth modeling, physiology

The study of the growth of microbial cultures has been a basic method of understanding bacterial physiology since the pioneering work of Jacob and Monod and the Copenhagen group in the 1950s to 1960s (1). Today, automated platforms equipped with multiwell plate readers can rapidly generate large sets of microbial growth data. Several computational tools have been developed for the rapid analysis and interpretation of these growth data sets (2–7). However, the growth of clinical isolates, fastidious organisms, or microbes under various stressors often generates curves that do not follow standard logistic or sigmoidal shape. Popular tools using classical mathematical models of growth, such as logistic or Gompertz equations, struggle to infer kinetic parameters for these microbial cultures. Nonparametric statistical approaches, including spline fitting, input estimation methods, or Gaussian process (GP) regression, are more effective at modeling the growth of atypical microbial cultures and estimating their growth parameters (8–11).

GP regression has especially shown tremendous potential for modeling growth curves. Unlike spline fitting, GP regression is robust to outliers and technical variation, can inherently pool replicates, reliably infer growth rates, and estimate growth parameters without cross-validation or bootstrapping (9, 10). While input estimation methods, such a regularized linear inversion, can also reliably infer growth rates (11), GP regression provides a principled framework for estimating uncertainty and statistical testing (10). Tonner et al.

Address correspondence to Firas S. Midani, Firas.Midani@bcm.edu.

* Present address: James Collins, Department of Microbiology and Immunology, School of Medicine, University of Louisville, Louisville, Kentucky, USA.

AMiGA is a general-purpose user-friendly software for the Analysis of Microbial Growth Assays with Gaussian Process regression

mSystems®

recently expanded their application of GP regression into a mixed-effects model of microbial growth and further illustrated how it can statistically estimate the impact of perturbations on growth dynamics (12). Both Swain et al. and Tonner et al. have openly shared tools for the analysis of microbial growth with GPs (9, 10, 12). However, these tools are either limited to measuring a very restricted set of growth parameters, are focused solely on statistical modeling of growth differences without inferring growth parameters, or are inaccessible to users who are unfamiliar with programming or Python. The wide adoption of the application of GPs for modeling microbial growth is thus hampered by the lack of a general-purpose and user-friendly software.

Here, we describe and showcase a new tool for the analysis of microbial growth assays (AMiGA) without any underlying assumptions about the shape of growth curves. AMiGA models growth curves with GP regression and infers biologically meaningful microbial growth parameters, including maximum specific growth rate (i.e., exponential growth rate), lag time, carrying capacity, and area under the curve (AUC). Because GPs do not assume any underlying shape for the input observations, we also show that AMiGA can quantify adaptation time, death, maximum death rate, and diauxic shifts. Finally, AMiGA can expand growth curves beyond time to include other experimental variables, for example, nutritional state (e.g., presence of substrate in culture), environmental conditions (e.g., pH status), microbial stressors (e.g., antibiotics), or phylogenetic identities (e.g., genotype). Users can accordingly test for functional differences in growth across distinct experimental conditions (10, 13). These statistical tests are agnostic to microbial growth parameters but rather detect differences between growth curves across all time measurements. Altogether, AMiGA streamlines various aspects of microbial growth data analysis, including quality control, data manipulation, growth curve fitting, and statistical testing.

To demonstrate the utility of AMiGA, we model the microbial growth of lab-adapted, clinical, and environmental isolates. We show how AMiGA can fit different growth dynamics, characterize diauxic shifts, describe Biolog phenotype microarray (PM) plates, and analyze standard growth assays. We also showcase how AMiGA-based testing for differential growth across distinct experimental conditions can extract useful insight from high-throughput growth assays.

## RESULTS

**Implementation.** AMiGA is an open-source, cross-platform Python package. Users interact with AMiGA via the command-line interface. We provide detailed tutorials to demystify the process for users with minimal background in using a command terminal. The main input to AMiGA is raw data files in text format, which are often exported by multiwell plate readers. Users can also pass metadata about each plate or each well using tables saved as text files. AMiGA can recognize Biolog PM plates based on file names and automatically assign wells to carbon, nitrogen, or phosphorous substrates. Users can optionally adjust many of the default parameters for analysis or visualization with a configuration file. AMiGA can analyze multiple data sets in a single batch, and it can pool biological and technical replicates before jointly modeling their growth curves.

**Estimating microbial growth parameters.** Using observed measurements of optical density (OD), we often want to describe the underlying function of growth often termed the growth curve. While preprocessing the data (see Materials and Methods), AMiGA transforms OD measurements with a natural logarithm (ln) then shifts measurements such that the first measurement is centered at zero. Using GP regression, AMiGA then infers the underlying growth curve, which quantifies microbial community size over time, and its first-order derivative, which quantifies the rate of growth over time (Fig. 1A and B). The algorithm then infers biologically meaningful growth parameters by either directly analyzing the growth and its derivative or by sampling from the posterior distribution of the predicted growth model. The sampling approach provides summary statistics for each of the growth parameters in terms of mean, standard deviation, and confidence interval. The estimated microbial growth parameters include classical growth characteristics, such as the carrying capacity ($K$; the maximum growth supported by the environment), area under the curve (AUC; the total growth supported by the environment over observed time), maximum

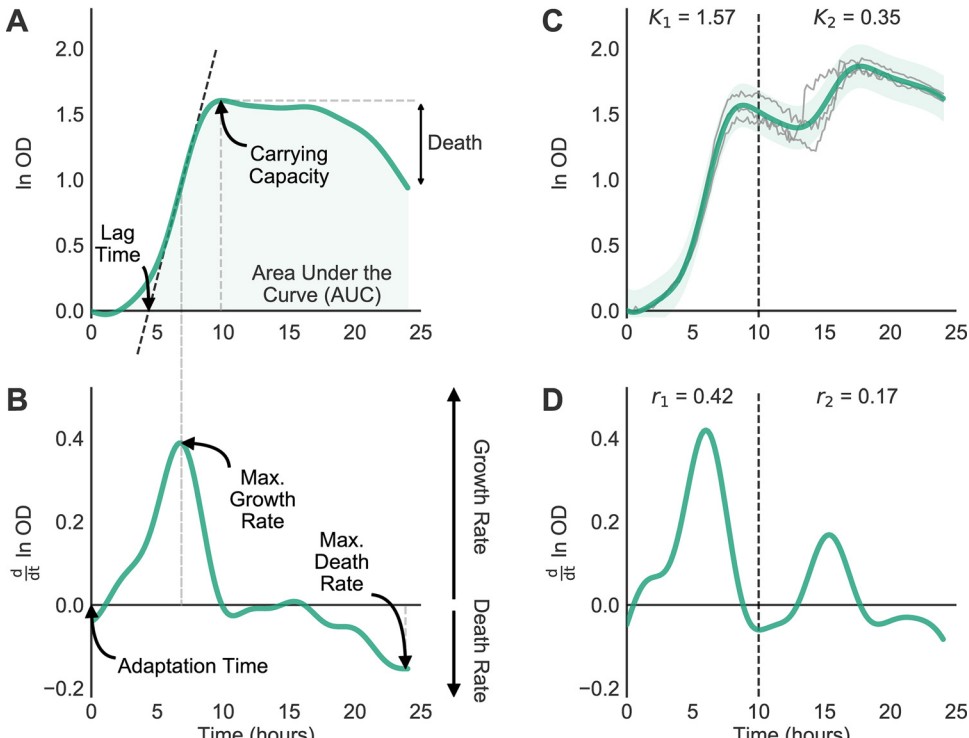

**FIG 1** AMiGA infers microbial growth parameters and characterizes diauxic shifts. (A, B) AMiGA predicted the mean growth and growth rate of a ribotype 053 *C. difficile* clinical isolate grown on 20 mM fructose as the primary carbon source using three technical replicates. Model estimates of the log-transformed growth and its derivative can be used to describe the following parameters: carrying capacity (*K*), area under the curve (AUC), maximum growth rate (*r*), maximum death rate, lag time, adaptation time, and stationary death. The black dashed oblique line indicates the tangent to the growth curve at maximum exponential growth. The gray dashed vertical lines map growth parameters to their respective time points, which are also exported by AMiGA. (C, D) AMiGA characterized the diauxic shift of a *C. difficile* clinical isolate of an unknown ribotype grown on minimal medium with 20 mM glucose as the primary carbon source using three technical replicates. AMiGA automatically detected two growth phases (separated by dashed vertical line), reaching maximum growth rates at 6.0 and 15.3 h, respectively. The inset text states the estimated carrying capacity, *K*, and maximum growth rate, *r*, for each unique phase. Bold green lines plot the predicted mean of the growth function or its derivative, and green bands indicate the predicted 95% confidence interval including measurement noise. The thin gray lines plot actual growth measurements.

growth rate (*r*; the maximum specific growth rate or exponential growth rate), doubling time (the time needed during exponential growth to double community size), and lag time (the time delay needed to initiate exponential growth, defined as the intersection of the tangent at maximum growth rate with the axis of time). In addition, AMiGA infers less commonly studied, but useful, growth characteristics, including maximum death rate (the most negative growth rate after reaching carrying capacity), death (the total loss of growth after reaching carrying capacity), and adaptation time (the time interval needed to reach a positive growth rate). Collectively, these growth parameters describe informative dynamics about microbial physiology under the specified experimental conditions.

**Characterizing diauxic shifts.** Diauxie is the biological phenomenon observed when a microbial culture undergoes two phases of growth (1). Diauxic growth often occurs when a microbial culture initially utilizes the most preferred carbon source in its environment but switches to a secondary source once the former is depleted (14). To identify diauxic shifts, we take advantage of a key feature of GPs, the derivative of a Gaussian process is also a Gaussian process. Therefore, we can easily infer the first- and second-order derivatives of each growth curve (9). Here, the first-order derivative estimates the growth rate over time, while the second-order derivative estimates the change in growth rate over time, which is a measure of the acceleration or deceleration of growth. Using these estimates, AMiGA then applies a novel iterative process and growth curve thresholds to detect

mSystems®

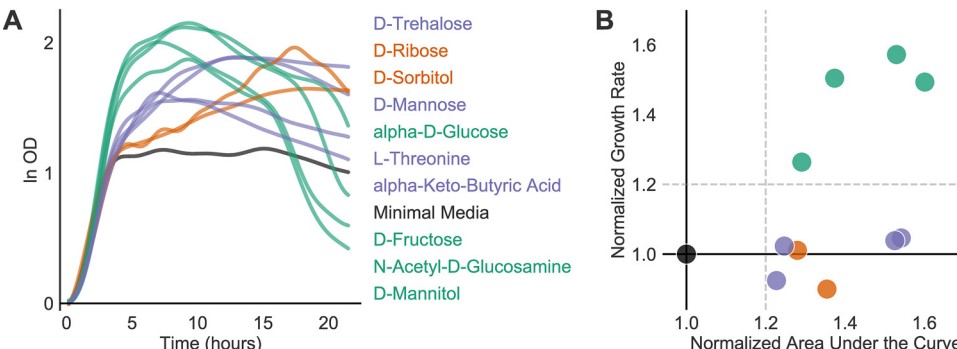

**FIG 2** *C. difficile* exhibited growth curves with various shapes that can be distinguished by growth parameters inferred by AMiGA. (A, B) CD2015, a ribotype 027 *C. difficile* isolate, was profiled with a Biolog phenotype microarray (PM1) plate with two technical replicates. Growth curves for each substrate were natural log transformed, baseline corrected, and then modeled jointly with GP regression by AMiGA. Growth parameters for all substrates were normalized to growth parameters on minimal medium by division. (A) CD2015 grew to a normalized AUC higher than 1.2 on 10 substrates. Substrate labels on the right are color coded and ordered by the final value of their corresponding growth curves. (B) On four substrates (green), CD2015 experienced growth rates that are at least 20% higher than growth rates on minimal medium. On ribose and sorbitol, *C. difficile* exhibited biphasic growth (orange). On remaining substrates (purple), *C. difficile* showed logistic or sigmoidal growth and grew at rates comparable to the rate of growth on the minimal medium (black). Dashed horizontal and vertical lines show arbitrary thresholds for color coding by normalized growth rate and calling positive growth, respectively.

diauxic shifts. Users can optimize parameters of this algorithm for their specific application in order to reduce false-positive calls (see Materials and Methods for technical details). This algorithm can detect diauxic shifts with rapid transition or short lag phase, as we see for one clinical isolate grown on minimal medium supplemented with glucose (Fig. 1C and D), or slow transition into secondary growth, as we see for a *C. difficile* clinical isolate grown on minimal medium (Fig. 2A; see also Fig. S1 in the supplemental material). Because GP regression does not assume any underlying shape of growth, this algorithm can capture two or more unique growth phases.

Here, multiple phases are considered unique if they are separated by a lag phase (defined as deceleration of growth rate to zero) and if the secondary phase results either in a substantial change of OD or reaches a sufficiently high growth rate. Users can select whether secondary shifts are called based on change in growth or growth rate and arbitrarily define the critical thresholds for these changes. For our *C. difficile* growth assays, we found that diauxic shifts are properly detected by a total change in OD during the secondary growth phase of at least 20% of the total change in OD during the primary growth phase.

**Modeling microbial growth dynamics.** *C. difficile* is a Gram-positive spore-forming pathogen that has recently become the most common hospital-associated infection in the developed world (15). *C. difficile* is a genetically diverse species, and distinct ribotypes are overrepresented in both human outbreaks and animals (16–18). Carbon substrate utilization by clinical *C. difficile* isolates demonstrates phenotypic diversity between ribotypes (19, 20). Here, we profiled CD2015, a ribotype 027 *C. difficile* clinical isolate, on a Biolog phenotype microarray plate (PM1) using two technical replicates. All wells in a Biolog PM plate are prearrayed with a single carbon source except for the first well, which lacks a carbon source and serves as a minimal medium control.

Using AMiGA, we pooled duplicate growth curves for each substrate, modeled them jointly with GP regression, and inferred microbial growth parameters (Data Set S1). To identify substrates that supported CD2015 growth, parameters were normalized by AMiGA to the minimal medium well. Predicted growth curves and parameters can be visualized with basic, but customizable, figures (Fig. S2 and S3). We detected significant growth (defined as a normalized AUC of at least 1.2) on 10 carbon substrates (Fig. 2). Because GP regression is a nonparametric approach, AMiGA can fit growth curves with various shapes (Fig. 2A). In minimal defined medium supplemented with a single carbon source, CD2015 exhibited multiple growth modalities, including rapid

growth followed by rapid death (green; normalized growth rate of $\geq$1.2), biphasic growth with a lower growth rate in the second phase for a prolonged period (orange; diauxie = true), and logistic or sigmoidal growth (purple and black; normalized growth rate of <1.2) (Fig. 2B and Fig. S4). As a bacterial generalist, *C. difficile* can colonize different nutritional niches in the gut (21). It is also capable of using amino acids as its sole energy and biomass source (22), which explains its ability to grow on minimal medium and may explain its biphasic growth on ribose and sorbitol. *C. difficile* may initially ferment amino acids via the Stickland pathway until they are depleted then transition to growth on sugars present in the environment (22, 23). However, it is less clear why rapid growth on certain monosaccharides was swiftly followed by rapid death.

A parametric approach for modeling growth would have poorly characterized some of these curves and misrepresented growth dynamics on different substrates. In addition, analysis that eschews fitting growth curves due to their atypical shapes and simply analyzes areas under the curve or carrying capacity would miss differences in growth dynamics. For example, CD2015 showed similar total growth on fructose and trehalose (95% confidence intervals for AUC, in units of ln OD $\times$ h, are 35.89 to 36.55 and 35.93 to 36.94, respectively), but its growth rates on these substrates were different (95% confidence intervals are 0.57 to 0.64 h$^{-1}$ and 0.33 to 0.48 h$^{-1}$, respectively). Importantly, our approach is not limited to modeling clinical isolates of *C. difficile* but can model growth dynamics of other microbes, including lab-adapted and environmental isolates, as we show for *Citrobacter sedlakii*, *Pseudomonas aeruginosa*, and *Yersinia enterocolitica* (Fig. S5). Because our nonparameteric approach models different growth shapes, it is especially useful in high-throughput screens for which manual validation of growth curves would be prohibitively laborious.

**Detecting differential growth by comparing growth parameters.** In our Biolog phenotyping, we noticed that certain monosaccharides promoted an atypical growth curve characterized by a rapid rise followed by a rapid decay in optical density. Substantial decay in optical density may be explained by autolysis due to rapid loss of energy-generating substrates (23), sugar-driven phage induction that leads to lysis (24), or smaller cell sizes due to reductive division or dwarfing (25). Because the substrate concentration in Biolog PM plates is proprietary information and thus unknown, we wanted to see if this phenomenon occurs at different concentrations of these substrates. We therefore assayed 11 clinical isolates of *C. difficile* representing four different ribotypes (a molecular classification of closely related strains) for their growth dynamics on minimal medium supplemented with either fructose or glucose as the sole carbon source (Table S1). We recapitulated the rapid growth followed by rapid death phenomenon at 20 mM concentrations of fructose and glucose (Fig. 3A). Higher concentration of 50 mM sugar did not result in the rapid decay experienced at a lower concentration within 24 h, although strains reached similar carrying capacity on both concentrations (Fig. 3A). Next, we estimated growth parameters and their 95% credible intervals on pooled experimental and technical replicates for each unique combination of ribotype, sugar, and substrate concentration (Fig. 3B). Growth on fructose exhibited more dramatic decay of OD after reaching carrying capacity. Indeed, death rates were also higher (more negative rates) at low concentrations of fructose than at low concentrations of glucose. In addition, differences in death rates between low and high concentrations were statistically significant for three ribotypes on fructose but only for one on glucose (Fig. 3B). In summary, low concentrations of simple sugars, such as glucose or fructose, can result in rapid lysis of *C. difficile* after reaching carrying capacity, and the rapid lysis was heightened for select ribotypes.

Ribotypes notably varied in the precision of their growth dynamics. For example, predicted growth curves for ribotype 027 strains had much wider confidence intervals than curves for ribotype 053 strains (Fig. 3B). The difference in confidence is due to differences in experimental variation across biological and technical replicates of each ribotype (Fig. S6). Difference in measurement variance is captured by the model-optimized Gaussian noise term, which is used in the estimation of confidence intervals (Fig. 3B). Here, noise is modeled as a single time-independent term and optimized for estimating measurement noise across all time points while maximizing model fit. To do so, it may, however, amplify confidence

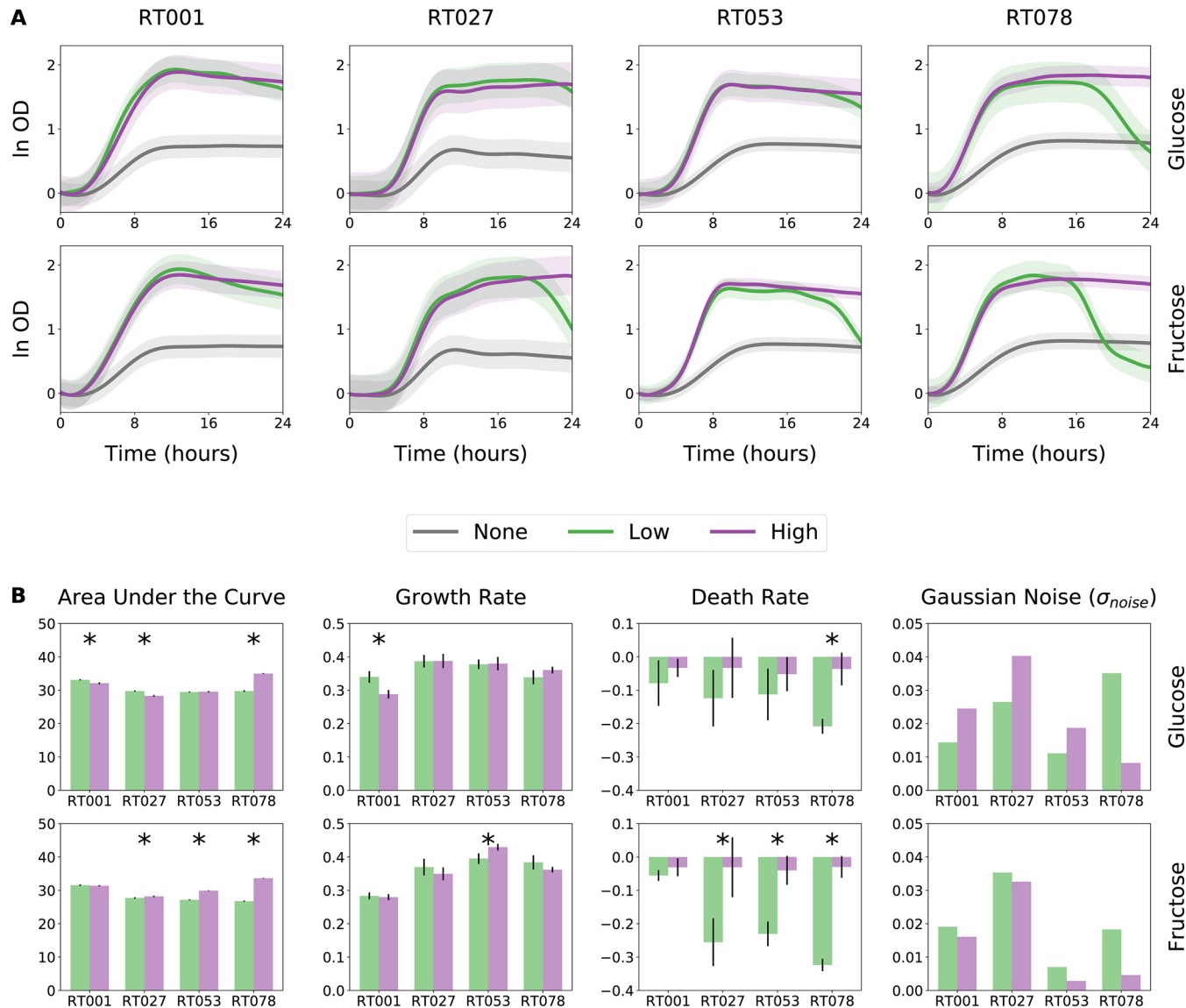

**FIG 3** Low concentration of glucose or fructose induce rapid death of microbial cultures in stationary phase. (A) AMiGA-predicted growth curves for clinical isolates belonging to four ribotypes (two RT001 isolates, four RT027 isolates, two RT053 isolates, and three RT078 isolates) grown on minimal medium supplemented with no (0 mM), low (20 mM), or high (50 mM) concentrations of either glucose or fructose. Growth for each isolate was measured with three technical replicates. Bold lines indicate the predicted mean of growth, and bands indicate the predicted 95% credible intervals, including measurement noise. (B) Summary of differences in growth using model estimates of the area under the curve, exponential growth rate, and stationary death rate. Sampling uncertainty was summarized with the model-estimated Gaussian noise. Error bars indicate the 95% credible interval, and asterisks indicate no overlap of credible intervals between low and high conditions for each ribotype and sugar combination.

intervals at time points where measurement variance is empirically low. For example, measurement noise is smaller during lag and exponential phases but larger during stationary and death phases. To fine-tune predicted confidence intervals, AMiGA users can opt to empirically estimate measurement noise as previously described (9), which can result in confidence intervals that more closely follow actual measurement noise over time (Fig. S6). Still, the time-independent noise term provides useful insight about our experiment. Here, we saw that ribotype 027 has the largest noise term and widest confidence intervals relative to other ribotypes. This may reflect larger phenotypic diversity among ribotype 027 isolates or may suggest a need for generating starting cultures in a manner that reduces variability in initial population size and physiology (26).

**Quantifying differential growth across all time points.** We initially contrasted growth on distinct experimental conditions by comparing growth parameters using

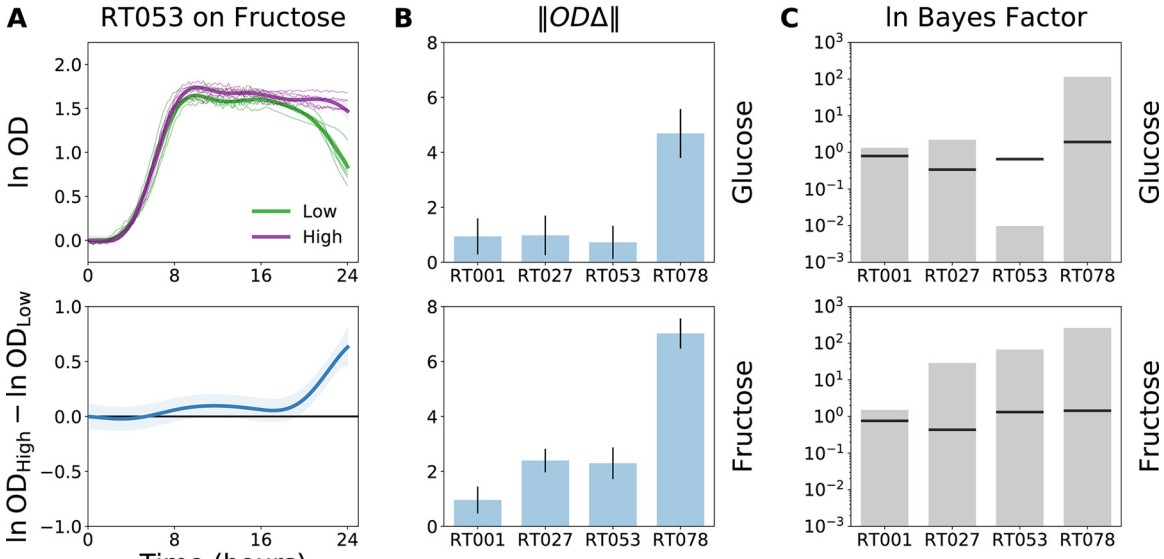

**FIG 4** Testing for differences in growth of each ribotype on low versus high concentrations of glucose or fructose. For each ribotype and sugar combination, we jointly modeled growth of microbial isolates on both low and high concentrations of each sugar. (A) Ribotype 053 *C. difficile* exhibited functional differences in growth on high versus low concentrations of fructose (OD$\Delta$ = ln OD$_{High}$ − ln OD$_{Low}$) with small differences in OD during early stationary phase and much larger differences in OD after 18 h. (B) Functional differences between growth on low versus high concentration of each sugar for each ribotype are summarized with the sum of functional differences, ‖OD$\Delta$‖, which quantifies the magnitude of differences between two curves. Error bars indicate 95% confidence intervals. (C) Log Bayes factor scores estimated how much the performance of these models improved by including the "concertation" covariate in addition to "time" in the models' input. Black horizontal lines indicate the FDR of ≤10% threshold based on 100 model permutations. Actual log Bayes factor scores above these thresholds are considered significant.

standard univariate statistical tests. This approach only detected differences captured by selected growth parameters. AMiGA can instead agnostically test for differential growth due to specific covariates (or conditions) as previously developed and described (10, 27). Briefly, the primary covariate of microbial growth measurements is the independent variable of time. We can extend a GP regression model to include additional categorical covariates that may contribute to differences in growth measurements over time, such as nutrient conditions (e.g., carbon substrate) or genotypes (e.g., *C. difficile* ribotype). We can then jointly predict microbial growth (ln OD) on these different conditions and the functional difference (OD$\Delta$ = OD$_{High}$ − OD$_{Low}$) in growth between these conditions (Fig. 4A and Table S2). Functional differences that deviate from zero suggest that different experimental conditions yield different growth dynamics. By aggregating these functional differences into a single metric, we can further compare how sugar concentration contributes to differences in growth based on *C. difficile* ribotype (Fig. 4B). AMiGA computed the Euclidean distance between two curves over time, which we refer to as the "sum of functional differences" (‖OD$\Delta$‖; see Materials and Methods for computation) (27). We confirmed that RT078 isolates experienced the largest growth differences due to sugar concentration, and that differences of growth due to fructose concentration for all ribotypes are higher than differences due to glucose concentration (Fig. 4B).

The sum of functional differences is a valuable metric for rank ordering conditions that impact overall growth dynamics. However, these summary scores are sensitive to the scale or magnitude of the growth curves. If the overall growth of *C. difficile* on both conditions is amplified by the same factor, then so are the functional differences between them. We can instead quantify the effect of an additional covariate on the growth model with less bias from the overall magnitude of growth using log Bayes factor scores (10). Briefly, growth is modeled on two hypotheses. The null hypothesis assumes that only time explains differences in growth data, while the alternative hypothesis includes additional covariates of interest in the model. We assess the

goodness of fit (or likelihood) of these competing models using the log Bayes factor score, which is the log ratio of the likelihood of the alternative hypothesis to the likelihood of the null hypothesis. Log Bayes factor scores higher than zero provide more evidence for the alternative hypothesis than for the null hypothesis. By permuting the covariate labels, we can also determine the false discovery rate (FDR) thresholds and only consider Bayes factor scores that outperform the threshold as significant.

We can broadly recapitulate our previous findings using the nonparametric log Bayes factor score. For the growth of each ribotype on either fructose or glucose, we computed a Bayes factor score, which evaluated if model predictions improve with knowledge of substrate concentration. Indeed, model performance improved for all conditions except for RT053 isolates grown on glucose, where the growth on low or high concentrations of glucose barely differed (Fig. 4C). Importantly, the log Bayes factor score highlighted differences in growth curves that are not captured by significant differences in death rate, such as the differential growth of RT001 and RT027 on distinct concentrations of glucose as well as the differential growth of RT001 on distinct concentrations of fructose (Fig. 3B). Although significant, these differences are minor in terms of functional differences or effect size, as indicated by the sum of functional differences (Fig. 4B).

As an additional example of AMiGA hypothesis testing, we reanalyzed the data set of Dunphy et al. (28) where they tested the growth of *Pseudomonas aeruginosa* transposon mutants on 20 mM *N*-acetyl-D-glucosamine. They found that seven out of eight mutants had significantly different carrying capacities compared with the ancestral strain. Our analysis further identified differential growth that can be missed by only analyzing carrying capacity (Fig. S7). The sums of functional differences, $\|OD\Delta\|$, indicated that all mutants exhibited statistically different growth compared with the ancestral strain. Most of these functional differences can be attributed to differences in carrying capacity, growth rate, and lag time. We confirmed that seven out of the eight strains grew to a higher OD or carrying capacity than the ancestral strain. We also showed that the eighth strain (PA14_57880) reached similar carrying capacity as the ancestral strain, but it did so with a shorter lag time and lower growth rate. We also showed that two strains with similar carrying capacities (PA14_41710 and PA14_44360) had significantly different lag time, growth rate, and total growth, as indicated by area under the curve. A comparison of these two isolates had a log Bayes factor score of 311.29 and a sums of functional differences of 1.36 (95% confidence interval of 1.04 to 1.68). Overall, differential testing can help users prioritize *post hoc* analysis and experimental validation by ranking conditions based on log Bayes factor scores or sums of functional differences.

## DISCUSSION

Recent advances in laboratory automation have dramatically increased the throughput of microbial growth assays. Important aspects of analyzing these data sets include screening for desired phenotypes (29), discovering genotype-phenotype relationships (27, 30), investigating cross talk of environmental pressures and microbial dynamics (28, 31), and predicting microbial fitness under a variety of conditions (32, 33). Yet, we are continuing to learn how growth dynamics are affected by a variety of technical and experimental factors (26). Scientists are also increasingly able to cultivate nontraditional or nonlab-adapted strains and manipulate them under a wide variety of treatments. The study of diverse microbes under widely different applications and growth conditions can thus generate growth modalities that do not follow standard sigmoidal growth and require more nuanced analysis. Here, we contribute a user-friendly software that can further streamline the analysis of microbial growth assays (AMiGA) with a nonparametric modeling approach.

We showcased our software using multiple examples to highlight several useful features. AMiGA can infer biologically meaningful parameters either by analyzing individual growth curves or pooling replicates of similar experimental conditions across multiple data sets. The latter approach enables inference with summary statistics for growth parameters, in particular the mean and confidence intervals. It can also probabilistically compare microbial growth under distinct conditions, which can aid scientists in ranking

experimental conditions for *post hoc* validation. We also developed a novel algorithm for the detection and characterization of multiphasic growth without any underlying assumptions of growth curve shape. Diauxie is a complex behavior characterized by two phases of growth that are often separated by a lag phase (1, 14, 34). There are no consensus formulations for diauxie, and current studies have relied on *ad hoc* analysis of growth data. Our novel algorithm for diauxie characterization can simplify its analysis and contribute to ongoing efforts for studying its behavior. Finally, emerging research on antibiotic- and phage-microbial interactions can benefit from automated approaches for measuring and quantifying the inhibition of microbial growth or death of microbial culture as we demonstrate here (29, 33). In summary, AMiGA streamlines the analysis and interpretation of growth curve assays in a high-throughput user-friendly manner and can be utilized in a wide variety of microbiology experiments.

## MATERIALS AND METHODS

**Bacterial strains and growth.** *C. difficile* strains were mostly clinical isolates obtained from the Michigan Department of Community Health (Table S1) (35). Growth assays were carried out at 37°C in anaerobic atmosphere (5% $CO_2$, 5% $H_2$, 90% $N_2$) using prereduced medium. Strains were cultured overnight in brain heart infusion (Difco) supplemented with 5% (wt/vol) yeast extract (BHIS). *C. difficile* cultures were diluted 1:10 in defined minimal medium (DMM) to a final OD of ~0.05 based on absorbance (620 nm) in a 1-cm cuvette with a spectrophotometer (Thermo Scientific, Genesys 20). DMM is the basal defined medium in Table 1 of Karasawa et al. (36) with several adjustments. The concentrations for 11 amino acids were lowered to the following: glycine, histidine, and tryptophan, 75 mg liter$^{-1}$; arginine, methionine, and threonine, 150 mg liter$^{-1}$; isoleucine and valine, 225 mg liter$^{-1}$; leucine, 300 mg liter$^{-1}$; cysteine, 400 mg liter$^{-1}$; and proline, 450 mg liter$^{-1}$; and the concentrations for biotin were increased to 0.125 mg liter$^{-1}$. Cultures were mixed 1:1 with sugar solutions for final concentrations of either 20 mM or 50 mM glucose or fructose. For Biolog PM assays, overnight growth of CD2015 was subcultured 1:5 in BHIS broth and grown to mid-exponential phase (OD ~0.6) then diluted in DMM to a final OD of 0.05 based on absorbance (620 nm) in a 1-cm cuvette. Each well in the prereduced Biolog PM plate was inoculated with 100 $\mu$l of final cell suspension. In remaining assays, cultures were grown in 200 $\mu$l volumes. All cultures were grown statically for 24 h and optical density (620 nm) was measured every 10 min immediately after 5 s of orbital shaking by a microplate reader (Tecan Life Sciences, Sunrise).

**Modeling growth data as a Gaussian process.** Microbial growth is defined as the observations of microbial abundance over time. Mathematical models, such as logistic or Gompertz equations, can describe a microbial growth curve (37). However, microbial growth does not always follow the standard sigmoidal shape. As a nonparameteric statistical approach, GPs can model microbial growth without making assumptions about the underlying shape or characteristics of its function. A GP is a probability distribution over functions (here, curves) where any finite number of observations of these functions, with independent and identically distributed Gaussian noise, are distributed as a multivariate normal distribution (38). In the GP framework, we assume that we have a set of observed functions $f(x) \in \mathbb{R}^N$ at input random vector of $x \in \mathbb{R}^N$, where $N$ is the number of random variables. The observed values are drawn from a multivariate normal distribution.

$$f(x) \sim \mathcal{N}(\mu(x),\ \Sigma)$$

The equation $\mu(x) \in \mathbb{R}^N$ is a mean function, and $\Sigma \in \mathbb{R}^{N \times N}$ is a covariance function or kernel where $\Sigma_{ij} = k(x_i, x_j)$ with $i, j \in 1 \cdots N$.

To model microbial growth curves, a GP can be specifically indexed at time such that a microbial growth function is a vector where each entry in the vector specifies a random variable, here, the function value $f(t_i)$ at a particular input time $t_i \in t_1, t_2, \cdots, T$.

$$\ln OD(t) \sim \mathcal{N}(\mu(t),\ \Sigma)$$

The natural log of optical density (ln*OD*) is now the observed function at input values of $t \in \mathbb{R}^T$, $\mu(t)$ is a mean function, $\Sigma$ is a covariance function or kernel, and $T$ indicates the number of observed random variables or time points.

While a Gaussian process is a distribution over an infinite number of arbitrary functions, we can bias a GP to infer functions that follow certain characteristics. To ensure the inference of smooth microbial growth curves, we can specify the priors of a GP to a mean function of zero, $\mu(t) = 0$, and the kernel or covariance function to a radial basis function (RBF) with a time-independent Gaussian noise hyperparameter.

$$k(t_i, t_j) = \sigma_{\mathrm{RBF}}^2 \exp\left(-\frac{(t_i - t_j)^2}{2\ell^2}\right) + \sigma_{\mathrm{noise}}^2 \cdot \delta_{i=j}$$

The RBF signal variance is $\sigma_{\mathrm{RBF}}^2$, $\ell^2$ is the RBF lengthscale, $\sigma_{\mathrm{noise}}^2$ is Gaussian noise hyperparameter, and $\delta_{i=j}$ is an indicator function that is equal to one when $i = j$ and zero otherwise.

In the above definition, the sampling uncertainty is modeled by the time-independent Gaussian noise hyperparameter $\sigma_{\mathrm{noise}}^2$. Users can opt to empirically estimate measurement noise, $\sigma_{\mathrm{emp}}^2$, and include it in

the kernel as previously described (9). In particular, we specify the kernel to be the sum of the RBF kernel and time-dependent noise parameters.

$$k(t_i, t_j) = \sigma_{\text{RBF}}^2 \exp\left(-\frac{(t_i - t_j)^2}{2\ell^2}\right) + \left(\sigma_{\text{noise}}^2 + \sigma_{\text{emp}}^2(t_i)\right) \cdot \delta_{i=j}$$

The empirical noise parameter, $\sigma_{\text{emp}}^2$, is estimated empirically using the variance across all replicates at each time point, which is then smoothed over time with a Gaussian filter (scipy.ndimage.gaussian_filter1d). Default width of the filter is 1 h but can be adjusted by users. Unlike model-estimated noise, empirically estimated noise is only used for model optimization and cannot be used for predicting OD at new time points.

**Estimating microbial growth parameters.** The GP model parameters are optimized to any given data by maximizing the marginal likelihood through integrating over all possible functions. The optimized hyperparameters can then be used to predict the latent or hidden function and sample new functions from its posterior distribution. Because the derivative of a GP is another GP (38, 39), we can also use GPs to make predictions about the derivatives of the growth curve (i.e., growth rate over time) and sample the posterior of the first and second derivatives of the GP.

AMiGA estimates growth parameters from the optimized model either directly by analyzing the latent function and its derivatives or by sampling many functions from the posterior of the latent function and its derivatives then estimating the growth parameters of each new sample. The latter approach provides a distribution for the estimates of each growth parameter, which are then summarized into their means, standard deviations, and confidence intervals.

The growth curve metrics of carrying capacity ($K$), maximum specific growth rate ($\mu_{\text{max}}$), and area under the curve ($AUC$) were estimated as previously described (10). Briefly, $K$ and $\mu_{\text{max}}$ are the maximum *a posteriori* (MAP) estimates of the growth, $\ln OD(t)$, and the growth rate, $(d/dt)\ln OD(t)$, functions. The estimate of $AUC$ was calculated as the Riemann sum of the $\ln OD(t)$ function. In particular, $\ln OD$ was predicted at evenly spaced time points, then linearly transformed with vector of time intervals, $a = \Delta t, \Delta t, \ldots$, such that an approximation of the AUC follows a normal distribution of $AUC \sim \mathcal{N}\left(a \cdot \mu, a \Sigma a^T\right)$, where $\mu$ and $\Sigma$ are the mean and covariance of the predicted $\ln OD(t)$ function.

AMiGA also computes additional growth parameters, including death, maximum death rate, adaptation time, lag time, and doubling time. Death is simply the absolute difference between the growth measurements at the final time point and the carrying capacity. The maximum death rate is estimated as the minimum of the negative of the derivative function $(d/dt)\ln OD(t)$, except $t$ is limited to $(t_K, \ldots, T)$ to capture maximum death rate after reaching carrying capacity at $t_K$, whereas $T$ indicates the final time measurement. Adaptation time is computed probabilistically as the time at which the 95% credible interval of the growth rate, $(d/dt)\ln OD(t)$, deviates from zero. Lag time is computed using its classical definition (37) as the intersection of the tangent line to the $(d/dt)\ln OD(t)$ function at maximum growth rate and the line parallel to the $x$ axis or time (9). Doubling time is computed as $\ln(2)$ times the inverse of the maximum specific growth rate.

**Detecting and characterizing multiple growth phases.** AMiGA applies a novel algorithm for detecting diauxic shifts that utilizes first- and second-order derivatives of growth measurements and a customizable heuristic for calling unique growth phases (see Fig. S1 in the supplemental material for illustration). AMiGA computes the first- and second-order derivatives of each latent function, which correspond to the growth rate and the change in growth rate over time, respectively. The second-order derivative indicates inflection points that are defined as $\left(d^2/dt^2\right)\ln OD(t) = 0$. Positive inflection points indicate acceleration of growth, while negative inflection points indicate deceleration. These inflection points also correspond to the valleys and peaks in the first-order derivative, respectively. Multiple phases are considered unique if they are separated by a lag phase, which is indicated by a negative inflection point in the second-order derivative (i.e., deceleration of growth rate to zero). Thus, each potential unique growth phase is bounded by two consecutive positive inflection points. AMiGA handles edge measurements by assuming positive inflection points at the first and last time points.

AMiGA then applies a heuristic to determine if each potential growth phase indicates a significant change in growth dynamics. A potential growth phase is deemed real if it either results in a substantial change of growth, $\ln OD(t)$, or reaches a sufficiently high growth rate, $(d/dt)\ln OD(t)$, relative to the primary growth phase. Users can select whether secondary shifts are called based on change in growth or growth rate and arbitrarily define the critical thresholds for these changes. For our *C. difficile* assays, we found that diauxic shifts are properly detected, with few false positives, by a total change in OD during the secondary growth phase of at least 20% of the total change in OD during the primary growth phase.

After computing the first-order and second-order derivatives, AMiGA iteratively compares potential phases based on their total growth (or growth rate). Starting with the phase with the smallest growth, AMiGA compares it to the primary phase with the largest growth. If the smallest phase is associated with growth that exceeds a certain ratio of the growth caused by the primary phase, it is confirmed as a unique growth phase. Otherwise, this nongrowth phase is merged to one of its adjacent phases. The nongrowth phase is merged to the adjacent phase with the lower activation energy, which is defined as the difference between the maximum growth rate during the nongrowth phase and the growth rates at either its left or right boundary (i.e., at its beginning and end). Growth phases are then reranked based on their total growth, and iteration continues. The iterative process is completed when the smallest growth phase is confirmed as a unique growth phase or when all potential phases have been merged into a single growth curve, thus indicating that no diauxic shifts (or shifts in growth phases) have occurred. Once all growth phases have been detected, AMiGA describes them by their growth parameters in addition to their time boundaries.

**Testing for differential growth.** AMiGA applies the framework of Tonner et al. (10) for Bayesian testing of differential growth between two conditions. A Gaussian process can be extended to model additional covariates or dimensions beyond time, where each dimension has a unique independent lengthscale. This is illustrated in vector notation of the squared exponential covariance as described in Solak et al. (39) where

$$k(x_i, x_j) = \sigma_{\text{RBF}}^2 \exp\left(-\frac{1}{2}(x_i - x_j)\ \Gamma (x_i - x_j)'\right) + \sigma_{\text{noise}}^2 \cdot \delta_{i=j}$$

Here, the random variables are now multidimensional with input vectors $x_i = [x_{i,1}, \cdots, x_{i,D}]$, $xi = [x_{j,1}, \cdots, x_{j,D}]$, $D$ is the number of dimensions and the first dimension of the random variable, $x_{i,1}$ or $x_{j,1}$ is always time, and the dimension-specific lengthscales are defined by $\Gamma = \text{diag}\left(\frac{1}{\ell_1^2}, \ldots, \frac{1}{\ell_D^2}\right)$, where $\ell_d^2$ is the RBF lengthscale for dimension $d$. Other model hyperparameters include $\sigma_{\text{RBF}}^2$ (RBF variance) and $\sigma_{\text{noise}}^2$ (Gaussian noise).

Accordingly, AMiGA tests for differential growth by comparing a null model, where the only dimension in input is time, $x_i = [\text{time}]$, with an alternative model where input is multidimensional, for example, $x_i = [\text{time, substrate}]$, with time as the first covariate and an additional covariate included as a binary variable $x_{i,2} \in \{0, 1\}$.

Differential growth is quantified with a Bayes factor score, defined as the ratio of the likelihood ($\mathcal{L}$) of the data given the alternative hypothesis or model ($M_1$) to the likelihood of the data given the null hypothesis or model ($M_0$).

$$\text{Bayes factor} = \frac{\mathcal{L}(\text{Data}|M_1)}{\mathcal{L}(\text{Data}|M_0)}$$

A Bayes factor score higher than 1 indicates stronger evidence for the alternative model than for the null model. High scores indicate that the additional covariate improves null model performance and suggest that growth curves are functionally different due to covariate effects. A false discovery rate (FDR) for each Bayes factor score is calculated by estimating the null Bayes factor score distribution. In particular, the label of the additional covariate was randomly assigned to each sample without replacement from the original distribution of the covariate in the model input. Then, a null Bayes factor score was calculated for each permutation of the model. The null distribution comprised 100 permutations of the data set, and FDR (default is ≤10%) was defined as the 90th percentile of the null Bayes factor distribution.

**Quantifying functional differences between conditions.** When testing for differential growth, AMiGA also models the functional difference due to the additional covariate. The functional difference across all time points, $OD\Delta(t)$, is defined as the difference in OD predicted for each condition (10) as follows:

$$OD\Delta(t) = \ln OD_A(t) - \ln OD_B(t)$$

Time is indicated by $t \in 1, \cdots, T$, and OD is predicted for conditions A and B jointly. These differences can be summarized into a single metric, $\|OD\Delta\|$, which is the Euclidean distance between the predicted OD curves and thus represents the magnitude of functional difference between conditions (27). We define the "sum of functional differences" as follows:

$$OD\Delta = \sqrt{\sum_{t=t_0}^{T} [OD\Delta(t)]^2}$$

Time is indicated by $t$, $t_0$ indicates the first time measurement, and $T$ indicates the final time measurements.

The uncertainty of the sum of functional differences is estimated by sampling 100 times from the posterior distribution of the functional difference, computing the sum of functional differences for each sample, then reporting the mean and 95% confidence interval of the distribution.

**Data preprocessing.** To account for background optical density and instrumentation noise and to prepare for growth modeling, AMiGA can apply several corrections in the following order. First, AMiGA can ignore the first few time points because plate reader measurements often exhibit abnormally high variation in OD at the beginning of a growth assay. Second, users can subtract blank medium controls from each well to account for background optical density due to medium and its variation over time. Third, growth curves belonging to group-specific control samples (e.g., minimal medium in Biolog PM) can be subtracted from treatment or target growth curves. Fourth, AMiGA handles negative or zero values by vertically shifting all measurements such that the lowest value is positive. The smallest possible vertical offset can be either defined by the user as the limit of detection of their plate reader assay or empirically estimated from the distribution of the change in OD between consecutive measurements. Fifth, growth data are transformed with a natural logarithm. Because the natural logarithm of zero equals one, changes in log OD over time thus indicate changes in OD relative to an arbitrary starting population size of one. Finally, the OD value at the first measurement can be estimated with polynomial regression of degree of five on the first five time points as previously described (10). Based on user preferences, polynomial-estimated OD measurement or the actual OD measurement at the first time point is then subtracted from all consecutive measurements in a curve. The only required steps in these corrections are

the handling of negative or zero values, natural logarithm transformation, and baseline correction to start growth curves at zero. The remaining steps are optional and can be requested and configured by users.

In this article, we eliminated the first measurement of each growth curve only for the CD2015 Biolog PM assay, did not correct for blank medium or control samples, and subtracted the first measurement after log transformation to start growth curves at zero. However, we encourage users to perform all necessary corrections in their assay as high background optical density or low signal-to-noise ratio in the early stages of growth can affect the inference of several growth parameters, including adaptation time, lag time, and growth rate.

Growth data sets for *C. sedlakii*, *P. aeruginosa*, and *Y. enterocolitica* were publicly available (see Data availability) and underwent minor preprocessing. Growth data for ancestral strain and transposon mutants of *P. aeruginosa* (Fig. S7) were based on four colonies per strain with three technical replicate curves per colony. We used only the median growth curve of technical replicates for each colony. Unlike other assays, the growth data for *Y. enterocolitica* (Fig. S5) were based on Biolog PM technology, where absorbance corresponded to colorimetric changes due to cellular respiration on nutrients in each well. The measurements for *Y. enterocolitica* were very noisy in the first few measurements (at the first time point, mean = 16.4, median = 15, minimum = 0, and maximum = 60), while the maximum absorbance detected across all time points and growth curves was 333 with an average of 126.2. Therefore, we forced a limit of detection of 20 units on all growth measurements of *Y. enterocolitica* to reduce the bias of the first few noisy time points on growth modeling.

**Data postprocessing.** AMiGA has options for exporting preprocessed and postprocessed data, including input and output of the GP model, predicted growth parameters, predicted mean and covariance for both growth and its first-order derivative, predicted Gaussian noise, $\sigma^2_{noise}$, as well as sampling uncertainty (i.e., measurement variance) if estimated empirically. Credible intervals for growth curves (or parameters) can be computed from the predicted means and standard deviations.

$$\ln OD(t) = \mu(t) \pm \Phi_t\left(1 - \frac{\alpha}{2}\right) \cdot \sigma(t)$$

Here, $\mu(t)$ and $\sigma(t)$ are the mean and standard deviation of the latent function, respectively, $\Phi_t$ is the inverse of the cumulative distribution function for the standard normal, and $\alpha$ indicates statistical significance, such that $(1 - \alpha)$ is the desired confidence level. The standard deviation for predicted OD at each time point is computed as the square root of the sum of variances due to RBF signal, $\sigma^2_{RBF}$, Gaussian noise, $\sigma^2_{noise}$, and empirically estimated measurement noise, $\sigma^2_{emp}$, as follows:

$$\sigma(t) = \sqrt{\sigma^2_{RBF} + \sigma^2_{noise} + \sigma^2_{emp}(t)}$$

Users can opt to plot confidence intervals and compute functional differences without accounting for Gaussian or measurement noise.

**Software implementation.** AMiGA is written in Python 3 (Python Software Foundation, https://www.python.org). It utilizes GPy for Gaussian process regression (40), Pandas (41), NumPy (42), and SciPy (43) for data manipulation and scientific computing, and Matplotlib (44) and Seaborn (45) for data visualization. Code for AMiGA and growth data for *C. difficile* analyzed in this article are available online (https://github.com/firasmidani/amiga) with detailed documentation and tutorials.

**Data availability.** Growth data for *C. difficile* analyzed in this article are available online; see https://github.com/firasmidani/amiga. Growth data for *C. sedlakii* (2), *Pseudomonas aeruginosa* (28), and *Yersinia enterocolitica* (7) were previously published and are available in public repositories listed in their corresponding articles. See https://github.com/dacuevas/PMAnalyzer for *C. sedlakii* data, https://github.com/lauradunphy/dunphy_yen_papin_supplement for *P. aeruginosa* data, and https://github.com/kevinVervier/CarboLogR for *Y. enterocolitica* data.

## SUPPLEMENTAL MATERIAL

Supplemental material is available online only.

**DATA SET S1**, XLSX file, 0.1 MB.
**FIG S1**, TIF file, 0.5 MB.
**FIG S2**, TIF file, 0.7 MB.
**FIG S3**, TIF file, 0.5 MB.
**FIG S4**, TIF file, 0.5 MB.
**FIG S5**, TIF file, 2.2 MB.
**FIG S6**, TIF file, 0.9 MB.
**FIG S7**, TIF file, 1.2 MB.
**TABLE S1**, DOCX file, 0.03 MB.
**TABLE S2**, DOCX file, 0.03 MB.

## ACKNOWLEDGMENTS

We thank Lei Pan (Baylor College of Medicine) for sharing Biolog PM1 data for CD2015, a ribotype 027 *C. difficile* isolate.

This work was supported by National Institutes of Health (NIH) grants to R.A.B. (U01AI124290 and R01AI123278). F.S.M. was supported by an NIH institutional training grant (T32DK007664). R.A.B. receives unrestricted research support from BioGaia AB, consults for Takeda and Probiotech, serves on the scientific advisory board of Tenza, and is a cofounder of Mikrovia.

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
