## [Reviewer comments · mSystems]

AMiGA: software for automated Analysis of Microbial Growth Assays

Firas Midani, James Collins, and Robert Britton

Corresponding Author(s): Firas Midani, Baylor College of Medicine

Review Timeline:

Submission Date:	April 26, 2021
Editorial Decision:	June 2, 2021
Revision Received:	June 7, 2021
Accepted:	June 16, 2021

Editor: Rafael Silva-Rocha

Reviewer(s): The reviewers have opted to remain anonymous.

Transaction Report:

DOI: <https://doi.org/10.1128/mSystems.00508-21>

June 2, 2021

Dr. Firas S Midani
Baylor College of Medicine
Molecular Virology and Microbiology
Houston, TX 77098

Re: mSystems00508-21 (AMiGA: software for automated Analysis of Microbial Growth Assays)

Dear Dr. Firas S Midani:

Thank you for submitting your manuscript to mSystems. We have completed our review and I am pleased to inform you that, in principle, we expect to accept it for publication in mSystems. However, acceptance will not be final until you have adequately addressed the reviewer comments.

Thank you for the privilege of reviewing your work. Below you will find instructions from the mSystemseitorial office and comments generated during the review.

Preparing Revision Guidelines

For complete guidelines on revision requirements, please see the Instructions to Authors at <https://msystems.asm.org/sites/default/files/additional-assets/mSys-ITA.pdf>. **Submissions of a paper that does not conform to mSystems guidelines will delay acceptance of your manuscript.**

Sincerely,

Rafael Silva-Rocha

Editor, mSystems

Journals Department
Reviewer comments:

Reviewer #1 (Comments for the Author):

In general terms, the authors have responded satisfactorily to my observations. However, there are still some minor comments and details that the authors should consider before accepting the manuscript.

-In Figure 2A the term Non-carbon control still appears

-The Materials and Methods section does not indicate where the data for *C. sedlakii* and *Y. Enterocolitica* come from. If it is mentioned that Growth data for ancestral strain and transposon mutants of *P. aeruginosa* were obtained from reference 28. This is indicated in the results section (line 315). In the materials section, it must be clear where all the data come from if they were not obtained experimentally.

L615: Data Availability? Where?

Reviewer #2 (Comments for the Author):

I previously reviewed this manuscript and the authors have thoroughly addressed all of my initial concerns, which were minor. The methodology, software, and documentation are a very important contribution and will be of broad interest.

Reviewer #3 (Comments for the Author):

The authors have appropriately taken into account most of my comments.

Reviewer comments (Author response in red):

Reviewer #1 (Comments for the Author):

In general terms, the authors have responded satisfactorily to my observations. However, there are still some minor comments and details that the authors should consider before accepting the manuscript.

-In Figure 2A the term Non-carbon control still appears

Thank you for catching this error. We have corrected the figure by changing “No-carbon Control” to “Minimal Media”.

-The Materials and Methods section does not indicate where the data for *C. sedlakii* and *Y. Enterocolitica* come from. If it is mentioned that Growth data for ancestral strain and transposon mutants of *P. aeruginosa* were obtained from reference 28. This is indicated in the results section (line 315). In the materials section, it must be clear where all the data come from if they were not obtained experimentally.

L615: Data Availability? Where?

We initially combined the Software Availability section and the Data Availability section as a single section titled “Software implementation and data availability”. For clarity, we have separated them into their own individual sections. Now, we also include links to the public repositories where the data for *C. sedlakii*, *P. aeruginosa*, and *Y. Enterocolitica* are available, in addition to citing their corresponding references.

Reviewer #2 (Comments for the Author):

I previously reviewed this manuscript and the authors have thoroughly addressed all of my initial concerns, which were minor. The methodology, software, and documentation are a very important contribution and will be of broad interest.

Reviewer #3 (Comments for the Author):

The authors have appropriately taken into account most of my comments.

Thank you to all reviewers. Your feedback greatly improved our manuscript.

June 16, 2021

Dr. Firas S Midani
Baylor College of Medicine
Molecular Virology and Microbiology
Houston, TX 77098

Re: mSystems00508-21R1 (AMiGA: software for automated Analysis of Microbial Growth Assays)

Dear Dr. Firas S Midani:

Your manuscript has been accepted, and I am forwarding it to the ASM Journals Department for publication. For your reference, ASM Journals' address is given below. Before it can be scheduled for publication, your manuscript will be checked by the mSystems senior production editor, Ellie Ghatineh, to make sure that all elements meet the technical requirements for publication. She will contact you if anything needs to be revised before copyediting and production can begin. Otherwise, you will be notified when your proofs are ready to be viewed.

We recognize that the video files can become quite large, and so to avoid quality loss ASM suggests sending the video file via <https://www.wetransfer.com/>. When you have a final version of the video and the still ready to share, please send it to Ellie Ghatineh at eghatineh@asmusa.org.

Sincerely,

Rafael Silva-Rocha
Editor, mSystems

Journals Department
Fig S6: Accept
Fig S1: Accept
Table S1: Accept
Data S1: Accept
Fig S2: Accept
Fig S5: Accept
Fig S7: Accept
Table S2: Accept
Fig S4: Accept
Fig S3: Accept